# Psychometric properties of the Persian version of the hospitalized older adults' dignity scale for measuring dignity during acute hospitalization

Alireza Mirzaei[1,2], Mobina Jamshidinia[3], Marzieh Jahani Sayad Noveiri[4], Reza Imashi[5], Roghayeh Yaghoobi Saghezchi[5], Mohammad Javad Jafari[1], Reza Nemati-Vakilabad[2]*

1 Students Research Committee, School of Nursing and Midwifery, Ardabil University of Medical Sciences, Ardabil, Iran, 2 Department of Emergency Nursing, School of Nursing and Midwifery, Ardabil University of Medical Sciences, Ardabil, Iran, 3 Student Research Committee, School of Nursing and Midwifery, Guilan University of Medical Sciences, Rasht, Iran, 4 Department of Medical-Surgical Nursing, School of Nursing and Midwifery, Guilan University of Medical Sciences, Rasht, Iran, 5 Department of Emergency Medicine, School of Medicine, Ardabil University of Medical Sciences, Ardabil, Iran

* nematireza1998@gmail.com

## Abstract

### Background

Providing dignified care is an essential aspect of high-quality nursing, and upholding patients' dignity is emphasized in the professional standards for nurses. This research evaluated the reliability and validity of the Persian version of the Dignity of the Elderly in Acute Hospitalization Scale (HOADS-P) for assessing dignity during acute hospitalization.

### Methods

The HOADS was translated into Persian using the forward-backward method. A panel of ten older adults and experts assessed face and content validity. Construct validity was established through Exploratory Factor Analysis (EFA) and Confirmatory Factor Analysis (CFA) involving 400 hospitalized older adults aged 60 years and older.

### Results

The EFA found that the HOADS-P exhibited five latent factors, explaining 63.15% of the overall variance. Also, these factors were confirmed by CFA (CMIN=123.872, DF=79, $p<0.001$, CMIN/DF=1.568, RMSEA=0.042, GFI=0.953, CFI=0.987, NFI=0.966, RFI=0.954, and PCFI=0.743). Additionally, the HOADS-P demonstrated strong reliability, as indicated by a Cronbach's alpha coefficient (α) of 0.946.

**Data availability statement:** All relevant data are within the manuscript and its Supporting Information files (Data.sav).

**Funding:** The author(s) received no specific funding for this work.

**Competing interests:** The authors declared that they have no competing interests.

## Conclusion

The Persian version of the Dignity of the Elderly in Acute Hospitalization Scale (HOADS-P) is a valid and reliable tool for assessing dignity during acute hospitalization in Iranian health service centers.

## Introduction

Safe and respectful nursing care is a basic concept for providing quality care to older adults. Due to their unique characteristics, older adults are more vulnerable than other groups, so we see an increase in the need for these people for physical and mental care. Based on the results of studies, the hospitalization rate of the elderly in hospitals is estimated to be 21% [1] to 75% [2,3]. It is essential that the aging of the population is one of the biggest global successes that can be a reason for medical advances and social, economic, political, and health achievements in any society [4]. According to a report by the World Health Organization, the global population of older adults is projected to reach two billion by 2050, with approximately 80% residing in less developed or developing countries [5]. In Iran, it is predicted that people over 60 will make up 21.7% of Iran's population by 2050 [6]. As individuals age, they tend to rely more heavily on healthcare services to address a wide range of age-related physical and mental health issues [7–9]. In addition to the necessity for social service and health therapy organizations to anticipate and accommodate these changes, it is crucial to take into account the unique physical and psychological characteristics of this population [10].

The hospitalization of older adults is a common occurrence worldwide and is often accompanied by significant levels of stress [11]. These stress-inducing factors can have a detrimental impact on the dignity of older adults [12]. Dignity, being a fundamental human need [13], is a multifaceted concept that poses challenges in its precise definition due to its lack of clarity. Nonetheless, this concept is increasingly recognized as a pivotal concern in the provision of care [14].

Dignity is an essential aspect of the human experience, closely linked to our perception of our worth and significance. It is deeply intertwined with our interactions with others and can be strengthened or weakened due to these connections. According to a study by researchers [15], dignity plays a vital role in shaping how we navigate our relationships with others and how we view ourselves. Due to their wealth of life experiences, older adults often prioritize preserving their dignity [16]. The World Health Organization recognizes the significance of dignity, with one of the United Nations' five ethical principles concerning older adults being dedicated specifically to this concept. Four essential qualities are universally acknowledged as integral to dignity: respect, independence, competence, and communication [2,17]. Essentially, dignity serves as a vital characteristic that keeps a person feeling vibrant and engaged with life.

Maintaining the dignity of older adults helps create a sense of success and adapt them to their illnesses or limitations. In this way, older adults consider themselves

responsible, feel useful and valuable, and realize that they still have There are privileges. Also, damage to dignity is regarded as a threat to a person's health, which initially causes emotional changes, fear, despair, anger, shame, and at the next stage, causes a sense of worthlessness, insecurity, loneliness, depression, indifference, and even suicide [18].

Studies have shown that the dignity of older adults is threatened due to illness and care needs [19,20], and preserving the dignity of older adults is one of the most essential principles that must be followed when caring for them. Since nurses do not receive specialized training to provide services to older adults, it seems that they do not have the necessary ability to recognize the care needs of older adults, so they care for this age group exactly like other age groups. To appropriately respond to the educational needs of nurses in caring for older adults, it is necessary to know the real needs of older adults [21]. Maintaining the dignity of elderly patients during acute hospitalization can be challenging [22]. Therefore, to improve the care of older adults, nurses should focus on what older adults want, not what they consider necessary [3].

To date, researchers have done a lot of qualitative and quantitative research related to this concept, but having a standard tool to investigate the violation of the dignity of older adults can be a standard way to quantitatively measure the patient's experience of the health care provided to them. A standard way to quantitatively measure patients' healthcare experience is through patient-reported outcome measures (PROMs) [23]. Considering that dignity as a dynamic concept is influenced by demographic, clinical, cultural, and religious factors [22,24], the criteria and tools approved in other cultures may not be suitable for use in Iran due to the difference in demographic characteristics [24].

Hospitalized Older Adults' Dignity Scale (HOADS) has been introduced as an innovative and valid tool to evaluate the dignity of older adults during acute hospitalization, and due to having strong psychometric properties, it can provide a standard method for assessing the dignity of older adults. Among the features of this tool, we can point out its completeness, simplicity, and clarity, as well as the need for little time to check it. Considering that the current status of the dignity of older adults in acute care units in medical education centers is unclear, using a standardized and localized tool to evaluate this concept from the perspective of older adults can help us analyze how this problem in Iran helps. Therefore, this study determined the psychometric properties of the Persian version of the Hospitalized Older Adults' Dignity Scale (HOADS-P) for measuring dignity during acute hospitalization.

## Materials and methods

### Design and setting

In this methodological study, we aimed to conduct a psychometric evaluation of the Persian version of the Hospitalized Older Adults Dignity Scale (HOADS-P) to measure patient-reported dignity during the acute hospitalization of older adults. The study was conducted in the medical, surgical, intensive care units, and short-stay wards of educational-therapeutic hospitals affiliated with Ardabil University of Medical Sciences in northwestern Iran from 8 July to 27 November 2024.

### Participants

The sample size was determined based on the recommendation in psychometric studies that the variable-to-participant ratio be 1:10 [25–27]. To ensure adequate statistical power for factor analysis, we recruited more than ten participants for each item of the HOADS-P. The data collection process involved using a convenience sampling method to gather responses from a total of 400 older adults aged 60 years and older, with 200 participants contributing data for the Exploratory Factor Analysis (EFA) and another 200 for the Confirmatory Factor Analysis (CFA). The inclusion criteria for this study mandated that participants be 60 years or older, have been hospitalized for a minimum of three days, provide written informed consent, express willingness to participate in the study, and be proficient in Persian. Exclusion criteria included patients with incomplete questionnaire responses and those who were clinically unstable due to pain or a cognitive condition affecting their communication.

### Instrument

**Baseline characteristics.** The questionnaire's first section included seven closed-ended questions about age, gender, marital status, education level, occupation, admitted ward, and length of hospitalization.

**Hospitalized Older Adults Dignity Scale (HOADS).** Fuseini et al. [22] developed the Hospitalized Older Adults' Dignity Scale (HOADS) as a self-reported scale comprising 15 items to evaluate the dignity of elderly patients during acute hospitalization. In this scale, participants provide ratings based on a 5-point Likert scale, ranging from 0 (not applicable to me) to 4 (always). The HOADS's five subscales are as follows: shared decision-making (3 items), healthcare professional-patient communication (3 items), patient autonomy (4 items), patient privacy (2 items), and respectful care (3 items). The total score for the HOADS ranges from 0 to 60. Higher scores indicate a greater dignity of elderly patients during acute hospitalization. The original version of the HOADS showed good reliability in terms of internal consistency, as indicated by Cronbach's alpha coefficient of 0.763. The assessment subscales also demonstrated acceptable internal consistency, with Cronbach's alpha values ranging from 0.474 to 0.718.

### Psychometric evaluation

The study consisted of three phases: the translation procedure and the validity and reliability evaluations. Across these phases, several steps were carried out, including translation, face validity, content validity, construct validity, convergent and discriminant validity, internal consistency reliability, and test-retest reliability.

**Translation procedure.** The translation process commenced with obtaining approval from Dr. Abdul-Ganiyu Fuseini to employ the HOADS [22]. Translating the scale into Persian adhered to the World Health Organization (WHO) guidelines [28] and employed the forward-backward method. The process involved two independent bilingual translators who translated the original scale into Persian, and their translations were rigorously compared to the original version by research team members. Following this, a standardized version was established. In the subsequent phase, a third professional translator, specializing in nursing concepts and unaware of the original scale, translated the Persian version into English. The developer then reviewed the translated version to confirm the accurate representation of the main concepts. Dr. Abdul-Ganiyu Fuseini provided feedback, which was integrated into the final Persian version of the HOADS. This version was then assessed for validity and reliability.

**Face validity.** The face validity of the HOADS-P was assessed using a qualitative approach. Through purposive sampling, the HOADS-P was administered to a group of 10 older adults aged 60 and above who were hospitalized at the time. Following completion of the initial assessment tool, cognitive interviews were carried out to collect detailed feedback on the relevance, difficulty, and ambiguity of the individual items. This step was crucial for ensuring that the assessment tool was effective and accurately captured the intended information [29]. Subsequently, revisions were implemented based on feedback, primarily focusing on clarifying any problematic expressions identified during the cognitive interviews.

**Content validity.** The content validity of the HOADS-P was evaluated using qualitative and quantitative methods. The same panel of ten experts—each holding a PhD in Gerontology or Nursing and selected via purposive sampling—participated in both the qualitative and quantitative assessments. For the qualitative evaluation, the experts conducted an in-depth examination of the instrument's content, providing feedback on grammar, wording, and overall clarity. Their insights were instrumental in refining the assessment.

The assessment of the HOAD-P involved a quantitative evaluation of the content validity ratio (CVR) and the content validity index (CVI). These measures were utilized to ensure that the content of the HOAD-P was relevant, accurate, and comprehensive. When calculating the CVR, experts were required to individually assess each item using a 4-point scale, ranging from 1 (not relevant) to 4 (highly relevant). A CVR value exceeding 0.62 was considered acceptable and suggested that the content was essential [30]. The experts used a 4-point Likert scale to assess each item's relevance, clarity, and simplicity for the CVI calculation. The scale ranged from 1, indicating "not relevant," to 4, indicating "completely relevant." A CVI value higher than 0.78 was deemed relevant and acceptable [31].

A comprehensive examination was conducted to detect any floor or ceiling effects that suggest potential problems with the HOAD-P's content validity. If the proportion of participants scoring at the lowest or highest possible values (floor or ceiling effects) exceeds 15% in the assessment results, it could suggest that the assessment tool lacks content validity, indicating that it may not be fully capturing the range of abilities or traits it was designed to measure [32].

## Construct validity

**Exploratory Factor Analysis (EFA).** The Kaiser-Meyer-Olkin (KMO) was conducted to assess the sampling adequacy for factor analysis. This statistical measure provides insight into whether the variables selected for the study are suitable for factor analysis [33]. Bartlett's test of sphericity is a statistical method utilized to evaluate the interrelatedness of observed variables in a factor analysis model. This test examines whether the correlations between variables are significantly different from zero. A KMO value greater than 0.60 and a significant Bartlett's test of sphericity value (< 0.05) were used as criteria for determining the suitability of the data for EFA [33]. After computing the correlation matrix between the variables, we performed factor extraction using Principal Component Analysis (PCA) with Varimax rotation. We thoroughly assessed the factor loading of each item in both the original factor matrix and the rotated matrix. It was essential for us to confirm that the factor loading of each item was at least 0.3 to ensure the reliability and validity of our results [34]. Notably, factors with an eigenvalue of 1 or higher and communalities of more than 0.2 were considered statistically significant [35].

**Confirmatory Factor Analysis (CFA).** In the subsequent stage, the factor structures derived from EFA were validated through Confirmatory Factor Analysis (CFA) using the second random dataset comprising 200 participants. The maximum likelihood estimation (MLE) method was used to determine the most probable values for the statistical model's parameters. The following model fit indices were used to assess the model fit: Minimum discrepancy divided by degrees of freedom (CMIN/DF) ≤ 3, Root Mean Square Error of Approximation (RMSEA) < 0.08, Goodness of Fit Index (GFI) > 0.90, Comparative Fit Index (CFI) > 0.90, Relative Fit Index (RFI) > 0.90, and Parsimony Comparative Fix Index (PCFI) > 0.50 [36,37]. Notably, factor loadings exceeding 0.3 were deemed satisfactory [38].

**Convergent and discriminant validity.** To assess the convergent validity of the HOADS-P, it's essential that the Composite Reliability (CR) exceeds 0.7 and the Average Variance Extracted (AVE) is greater than 0.5 for each construct. According to Fornell and Larcker, if the AVE is less than 0.5 but the CR is greater than 0.7, the convergent validity for constructs can still be considered acceptable [39].

In our research, we analyzed discriminant validity using the Heterotrait-Monotrait (HTMT) correlation ratio developed by Henseler et al. [40]. To establish discriminant validity, it is crucial to confirm that the values present in the HTMT matrix are below the threshold of 0.9 [40]. This analysis helps ensure that the measures used in the study are distinct from one another and do not measure the same underlying construct.

**Reliability.** The reliability of the HOADS-P was evaluated using two reliability measures: internal consistency, which evaluates the extent to which all the items in the test are measuring the same construct, and test-retest reliability, which assesses the consistency of the results when the test is administered to the same individuals on two separate occasions [41]. The internal consistency of the HOADS-P was assessed through various statistical measures, including Cronbach's alpha ($\alpha$), McDonald's omega ($\omega$), Mean Inter-Item Correlation ($\rho$), and Coefficient H (H). For acceptable internal consistency, $\alpha$, $\omega$, and H values greater than 0.7 [42–47] and $\rho$ values between 0.15 and 0.5 were interpreted [48].

The test-retest reliability was assessed using the Intraclass Correlation Coefficient (ICC) with an absolute-agreement, two-way mixed-effects model. The stability of the HOADS-P was evaluated by collecting data from 50 older adults through simple random sampling at a two-week interval. The interpretation of the ICC was based on the following criteria: poor reliability (ICC < 0.5), moderate reliability (0.5 < ICC < 0.75), good reliability (0.75 < ICC < 0.9), and excellent reliability (ICC > 0.90) [49].

## Data analysis

The dataset underwent descriptive statistical analyses to summarize participants' baseline characteristics and EFA using IBM SPSS Statistics for Windows, version 24.0 (IBM Corp., Armonk, NY, USA). CFA was also conducted using AMOS Graphics, version 24.0. The statistical threshold for determining significance was established at $p < 0.05$.

## Ethical considerations

The research conducted for this study received approval from the Research Ethics Committee of Ardabil University of Medical Sciences under Approval ID: IR.ARUMS.REC.1403.088. The study followed the principles outlined in the Declaration of Helsinki (1975) for medical research. Additionally, the researchers provided a detailed explanation of the study's nature and objectives to the participants. The participants were informed that participation in the survey was voluntary and required written informed consent. They were also assured that they could withdraw from the study at any time. Furthermore, the results were published anonymously to ensure the confidentiality of the data.

## Results

### Characteristics of the participants

In total, 400 older adults who were admitted to educational-therapeutic centers participated in this study. The mean age of the participants was 66.2 years, with a standard deviation of 4.16. Nearly half of the participants were male ($n = 217$, 54.3%), and the majority were admitted to medical units ($n = 172$, 43.0%). Participants' characteristics are presented in **Table 1**.

### Face validity

After incorporating feedback from older adults, we conducted a comprehensive review of the material and implemented minor modifications to enhance linguistic precision and overall clarity. Consequently, we have opted to maintain the existing items in their current state.

### Content validity

The study's results showed that every item assessed had a CVR value higher than 0.62, ranging from 0.7 to 1. Furthermore, the CVI scores for each item were all above 0.78, ranging from 0.8 to 1. During the qualitative phase of content validity, the expert panel unanimously agreed that the grammar, phrasing, and item allocation about the specific factor were suitable for the research objectives.

Also, the results indicated that none of the participants scored at the lowest or highest possible level, meaning no floor or ceiling effects were observed. Specifically, less than 15% of the participants achieved the lowest or highest possible score for all items. These findings prove that the HOADS-P is reliable and valid for assessing the dignity of elderly patients during acute hospitalization.

### Construct validity

**Exploratory Factor Analysis (EFA).** The Kaiser-Meyer-Olkin (KMO) measure of sampling adequacy was 0.811, indicating that the sample size was sufficient for factor analysis. Additionally, Bartlett's test of sphericity revealed a statistically significant result ($x^2 = 998.39$, df = 156, $p < 0.001$). This indicates that the correlation matrix is appropriate for conducting factor analysis. Through PCA with Varimax rotation, five latent factors with eigenvalues > 1.0 were extracted, explaining 63.15% of the total variance in care-related regret intensity among clinical nurses (**Table 2**).

**Confirmatory Factor Analysis (CFA).** After extracting the structure, we assessed it using CFA in the subsequent step. When using the five-latent factors model, the CFA results indicated that all of the items in the model showed a factor loading of 0.3 or higher (ranging from 0.73 to 0.94) (**Fig 1**). The model's fit was further evaluated by examining goodness-of-fit indices.

**Table 1. Characteristics of the participants (*n* = 400).**

| Variable | Categories | Mean ± SD | |
|---|---|---|---|
| Age (years) | | 66.23 ± 4.16 | |
| Length of hospitalization (days) | | 6.21 ± 3.53 | |
| | | **No.** | **Percentage** |
| Gender | Male | 217 | 54.3 |
| | Female | 183 | 45.8 |
| Marital status | Married | 194 | 48.5 |
| | Single | 34 | 8.5 |
| | Divorced/widowed | 172 | 43.0 |
| Level of education | No formal education | 55 | 13.8 |
| | Basic education | 113 | 28.2 |
| | Secondary education | 206 | 51.5 |
| | Tertiary education | 26 | 6.5 |
| Occupation | Unemployed | 99 | 24.8 |
| | Employed | 50 | 12.5 |
| | Self-employed | 64 | 16.0 |
| | Retired | 187 | 46.8 |
| Ward admitted | Emergency | 64 | 16.0 |
| | ICU | 20 | 5.0 |
| | Medical | 172 | 43.0 |
| | Surgical | 144 | 36.0 |

Abbreviations: SD, Standard deviation.

All the indices, including CMIN = 123.872, DF = 79, $p < 0.001$, CMIN/DF = 1.568, RMSEA = 0.042, GFI = 0.953, CFI = 0.987, NFI = 0.966, RFI = 0.954, and PCFI = 0.743, confirmed an excellent fit for the final model of the HOADS-P.

## Convergent and discriminant validity

The results showed that the AVE for the five latent factors of the HOADS-P exceeded 0.5, indicating strong convergent validity. Furthermore, the CR for the latent factors of the HOADS-P was above 0.7, suggesting the successful establishment of convergent validity for all constructs. Regarding discriminant validity, the HTMT ratio results revealed that the correlation among the five latent factors (ranging from 0.366 to 0.542) was less than 0.90, demonstrating strong discriminant validity for all constructs (**Table 3**).

## Reliability

The reliability of all constructs was assessed, and high internal consistency was found. This was evidenced by Cronbach's alpha, McDonald's omega, and Coefficient H values, all of which exceeded 0.7. Furthermore, the mean inter-item correlation values fell within the range of 0.15 to 0.5, indicating strong consistency and reliability among the constructs.

The stability of the HOADS-P was assessed using the ICC. The final model's overall ICC value of 0.942 (95% CI, 0.933–0.951) indicated a high level of stability (**Table 3**).

## Discussion

How older adults are cared for in the hospital is crucial for their overall well-being and recovery. As the global population ages, it's increasingly important to understand and address their specific needs in hospitals. Despite the widespread agreement that dignity is a fundamental human right, there is currently no standardized method for measuring

**Table 2. The result of exploratory factor analysis of the HOADS-P ($n = 200$).**

| Factor | Items | Factor loading | $h^2$ | λ | % of variance |
|---|---|---|---|---|---|
| Shared decision-making | 1. HCPs include me in discussions about my care. | 0.813 | 0.757 | 3.146 | 48.28% |
| | 2. HCPs involve me in decisions about my care. | 0.842 | 0.798 | | |
| | 3. HCPs respect my choices about my care. | 0.879 | 0.799 | | |
| HCP-patient communication | 4. HCPs provide me with adequate information about my health condition. | 0.712 | 0.694 | 3.225 | 51.09% |
| | 5. HCPs provide me with adequate information about my treatment. | 0.821 | 0.788 | | |
| | 6. HCPs provide me with adequate information about my medications. | 0.848 | 0.781 | | |
| Patient autonomy | 7. HCPs pay attention to me when I speak. | 0.847 | 0.754 | 2.957 | 46.47% |
| | 8. HCPs seek my permission before performing any procedures on me. | 0.775 | 0.685 | | |
| | 9. HCPs respect my choice for my family to be involved in my care (e.g., bathing and toileting). | 0.736 | 0.679 | | |
| | 10. HCPs provide satisfactory assistance when I need help. | 0.806 | 0.775 | | |
| Patient privacy | 11. HCPs provide privacy when discussing issues about me. | 0.664 | 0.621 | 3.820 | 47.01% |
| | 12. HCPs provide privacy when providing care. | 0.702 | 0.653 | | |
| Respectful care | 13. HCPs treat me with compassion. | 0.723 | 0.696 | 3.619 | 50.75% |
| | 14. HCPs show respect for my religious beliefs. | 0.801 | 0.781 | | |
| | 15. HCPs respond to my needs in a timely way. | 0.828 | 0.782 | | |

Abbreviations: $h^2$, Communalities; λ, Eigenvalue.

how well the dignity of older adults is maintained during hospitalization. This study collaborated with others to assess the Hospitalized Older Adults' Dignity Scale (HOADS) tool in Persian. The objective was to determine the reliability and validity of this tool, which would enable healthcare personnel to assess and enhance the level of dignity provided to elderly patients in hospitals. Implementing this tool could lead to more patient-centered care and improved outcomes.

Establishing the face validity of the HOADS-P involved a collaborative effort that actively sought and incorporated feedback from older adults. Their valuable insights and perspectives led to the incorporation of minor modifications that notably enhanced the tool's linguistic clarity without compromising the integrity of its core items. This iterative process underscores an essential principle – the importance of participant input in developing assessment tools that accurately capture their unique experiences and needs [50]. This collaboration not only enriches the quality of the tool but also ensures that it reflects the diverse perspectives of the older adult population. Furthermore, the unanimous agreement among experts regarding the grammar, phrasing, and allocation of items significantly validates of the instrument's content. It affirms the tool's linguistic and structural integrity and reinforces its credibility as a valid and reliable assessment instrument, instilling user confidence [51]. In a study by Abdul-Ganiyu Fuseini et al., they developed and validated the HOADS using the "Guide to Best Practices for the Development and Validation of Scales for Health, Social, and Behavioral Research." The PROM (Patient-Reported Outcome Measure) met eight criteria outlined in the manual, including domain identification and item generation, content validity, face validity, pretesting, construct validity, convergent validity, internal consistency reliability, and test-retest reliability [22].

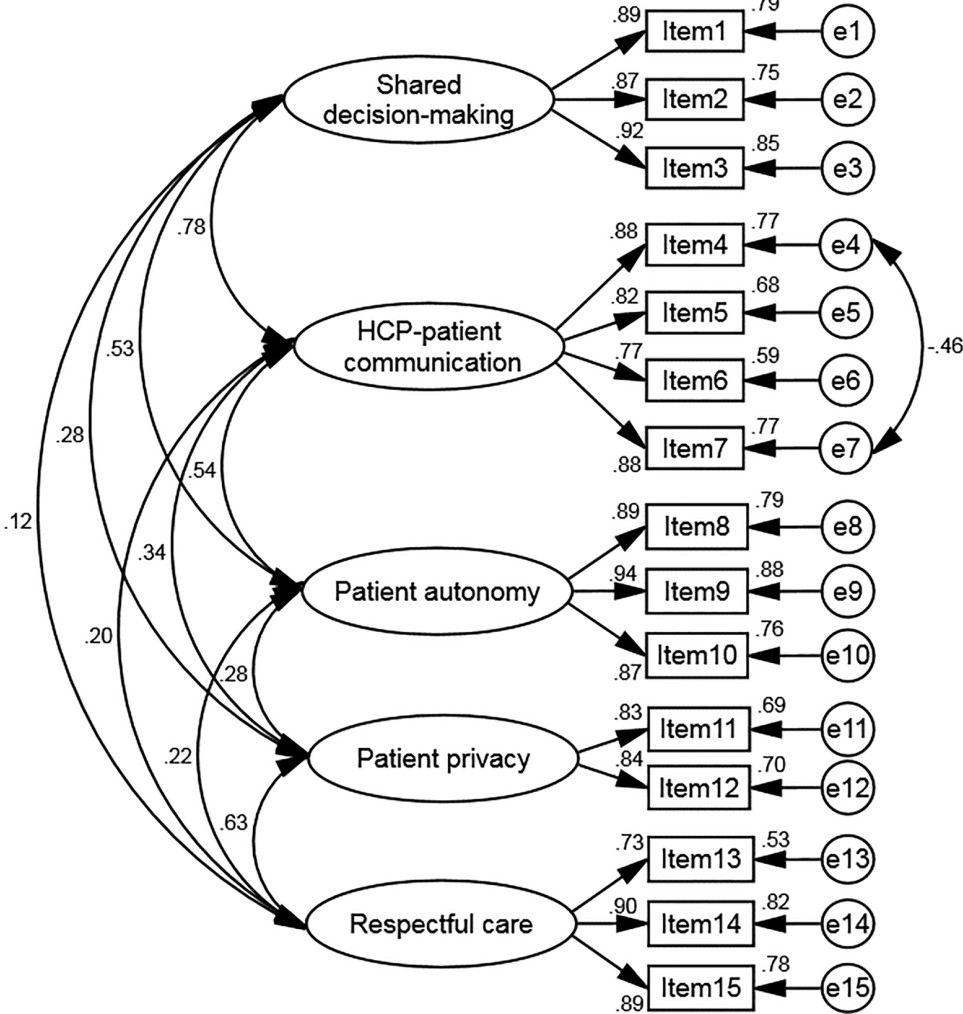

**Fig 1. CFA and the standardized parameter estimates of the modified model of the HOADS-P.**

**Table 3. The results of the convergent validity and reliability of the HOADS-P (*n* = 200).**

| Latent factors | AVE | CR | α | ω | H | ρ | ICC (95% CI) |
|---|---|---|---|---|---|---|---|
| Shared decision-making | 0.798 | 0.922 | 0.798 | 0.807 | 0.925 | 0.467 | 0.777 (0.711–0.826) |
| HCP-patient communication | 0.703 | 0.904 | 0.874 | 0.881 | 0.912 | 0.498 | 0.873 (0.847–0.896) |
| Patient autonomy | 0.841 | 0.927 | 0.890 | 0.893 | 0.937 | 0.473 | 0.886 (0.863–0.906) |
| Patient privacy | 0.697 | 0.827 | 0.872 | 0.875 | 0.821 | 0.496 | 0.865 (0.832–0.891) |
| Respectful care | 0.711 | 0.926 | 0.862 | 0.921 | 0.902 | 0.476 | 0.860 (0.832–0.885) |
| Total (15-item HOADS-P) | – | – | 0.946 | 0.945 | 0.961 | 0.468 | 0.942 (0.933–0.951) |

Abbreviations: AVE, Average Variance Extracted; CR, Composite Reliability; α, Cronbach's Alpha; ω, McDonald's Omega; H, Coefficient H; ρ, Mean Inter-Item Correlation; ICC, Intraclass Correlation Coefficient; CI, Confidence Interval.

Our study demonstrated robust results regarding content validity, with CVR values exceeding the 0.62 threshold and CVI scores above 0.78. These robust results should instill confidence in the study's findings. Following a thorough evaluation, experts rated the relevance of each item of the PROM using a 4-point scale, with the calculated Item Content Validity Index (I-CVI) indicating strong alignment with the measure of dignity for hospitalized older adults. Specifically, items with I-CVI values of 0.80 were deemed to have excellent content validity [22,52]. Additionally, the absence of floor or ceiling effects, indicated by the fact that less than 15% of participants scored at the extremes, suggests that the HOADS-P effectively captures a range of dignity-related experiences during acute hospitalization. The careful consideration of expert feedback, including proposed adjustments for items rated between 1 and 3, further enhances the tool's relevance and applicability. The evaluation criteria established for content validity ensured that only items with robust scores remained, reinforcing the integrity of the PROM. This reliability and validity are both adequate and robust, critical for ensuring that the tool can be used confidently in clinical settings to assess the dignity of elderly patients. The S-CVI/Ave, calculated with a criterion of 0.90 as the lower limit of acceptability, substantiates the overall effectiveness of the PROM in capturing the nuanced experiences of dignity among older adults during hospitalization [22,52].

The Exploratory Factor Analysis (EFA) revealed a KMO measure of 0.811, indicating that the sample size used in the study is suitable for factor analysis. This high KMO value is significant in social and clinical research, as it demonstrates adequate correlations among variables, which is essential for extracting latent factors. Additionally, the significant Bartlett's test of sphericity ($\chi^2 = 998.39$, $p < 0.001$) confirmed the appropriateness of the correlation matrix for factor analysis. The extraction of five latent factors, which explain 63.15% of the variance in care-related regret intensity among clinical nurses, suggests that the HOADS-P is a comprehensive tool that effectively captures multiple dimensions of patient experience. This reassures us of the HOADS-P's role in understanding nurses' emotions and experiences when facing caregiving challenges, ultimately improving the quality of healthcare services. In a study conducted by Abdul-Ganiyu Fuseini et al., the KMO measure of sampling adequacy was reported at 0.736, further supporting the adequacy of the sample size for factor analysis [22].

Moreover, their findings from Bartlett's test of sphericity yielded a significant chi-square value of 996.203 with 153 degrees of freedom ($p < 0.001$), indicating that the variables under consideration were well-suited for factor analysis [22]. The results demonstrate the effectiveness of factor analysis in evaluating complex concepts in healthcare and affirm the reliability of tools like the HOADS-P in capturing patient experiences. These findings offer a promising approach to improve public health outcomes and support healthcare professionals in enhancing care quality and patient satisfaction [53].

To understand the underlying structure of our measurement tool, we first conducted an EFA. After that, we performed a CFA to confirm a proposed five-latent factor model. All items showed factor loadings above 0.3 during the analysis, supporting the selected model [54]. The model fit indices, including a Root Mean Square Error of Approximation (RMSEA) of 0.042 and a Comparative Fit Index (CFI) of 0.987, indicate a strong fit for the model, confirming the structural integrity of the HOADS-P instrument. This analysis validates the dimensionality of the HOADS-P instrument and its potential for use in various clinical settings. The high CFI value suggests that the model explains significant variance in the data. Additionally, the low RMSEA indicates minimal error in the approximation of the population covariance matrix, enhancing the reliability and validity of the instrument and making it a valuable tool for practitioners. Further research can explore the model's applicability across different populations and settings to confirm its usefulness.

The results of the convergent and discriminant validity analyses indicated that the average variance extracted (AVE) for the five latent factors surpassed the widely accepted threshold of 0.5, providing strong evidence of convergent validity. Moreover, the composite reliability (CR) values exceeding 0.7 further substantiated this assertion, suggesting that the constructs are accurately defined and reliably measured [55]. This reliability of the measurement model is crucial for the validity of the study's findings and the trustworthiness of the research [56]. The establishment of discriminant validity was also evident, as indicated by the Heterotrait-Monotrait (HTMT) ratios falling below the recommended threshold of 0.90.

This finding illustrates that the latent factors effectively capture distinct and unrelated aspects of the patient experience [57]. Based on the findings, the HOADS-P instrument demonstrates strong convergent and discriminant validity, making it suitable for assessing various dimensions of patient experience in clinical settings. Further studies should explore its effectiveness across different populations to verify its applicability.

The reliability assessments indicate strong internal consistency across all factors. This is supported by the high values of Cronbach's alpha, McDonald's omega, and Coefficient H, all exceeding 0.7, which indicates a reliable measurement tool [58]. The strong agreement and coherence among the measured factors, as suggested by the average inter-item correlation values ranging from 0.15 to 0.5, further reinforce the tool's accuracy [59]. The test-retest reliability of the HOADS-P is indicated by the Intraclass Correlation Coefficient (ICC) value of 0.942, showing its stability over time [60]. The tool can reliably assess dignity in different situations and at various times. This discovery highlights its potential for long-term studies and demonstrates its adaptability.

This study has several strengths, including the comprehensive validation of the Persian version of the Hospitalized Older Adults' Dignity Scale (HOADS-P) and the use of a diverse sample of 400 participants, which enhances the generalizability of findings. However, certain limitations should be acknowledged. The cross-sectional design limits our ability to examine changes in dignity over time, and the focus on a single cultural context may restrict the applicability of results to other settings. Furthermore, the self-reported nature of the instrument may be subject to cultural biases, as perceptions of dignity vary across cultures, and participants might underreport their true concerns due to factors like social desirability bias. There is also potential for response bias in reporting dignity levels, which could affect the accuracy of results. To address these limitations, future studies should employ longitudinal designs to track dignity changes during hospitalization, validate the instrument across different cultural contexts, and utilize mixed-methods approaches with more diverse populations to gain a more comprehensive understanding of the cultural and contextual factors influencing dignity perceptions.

## Conclusion

The results of this study highlight the importance of the HOADS-P as a reliable and valid tool for assessing dignity in elderly patients during acute hospitalization. Its strong psychometric properties, demonstrated through rigorous validity and reliability testing, confirm that the HOADS-P effectively captures the multidimensional nature of dignity in clinical settings. By implementing this scale, healthcare providers can gain meaningful insights into older adults' experiences, facilitating targeted interventions to uphold dignity and improve care quality. Given the growing elderly population, tools like the HOADS-P are essential for advancing patient-centered care that prioritizes dignity, ultimately enhancing health outcomes and patient satisfaction. Future research should expand beyond acute care settings to examine the scale's applicability in long-term care facilities, home healthcare, and rehabilitation centers, where dignity preservation is equally critical. Additionally, studies exploring the HOADS-P's utility in cross-cultural comparisons and integration into dignity-focused interventions would further strengthen its clinical relevance.

## Supporting information

**S1 File. Persian version of the HOADS.**
(DOCX)

**S2 File. Original version of the HOADS.**
(DOCX)

**S3 File. Data.**
(XLSX)

## Acknowledgments

The authors would like to thank all the older adults and the officials of the hospitals affiliated with the Ardabil University of Medical Sciences, the Student Research Committee of Ardabil University of Medical Sciences, the Vice Chancellor for Research of Ardabil University of Medical Sciences, and all those who helped us in this study.

## Author contributions

**Conceptualization:** Alireza Mirzaei, Marzieh Jahani Sayad Noveiri, Roghayeh Yaghoobi Saghezchi, Mohammad Javad Jafari, Reza Nemati-Vakilabad.

**Data curation:** Alireza Mirzaei, Mobina Jamshidinia, Marzieh Jahani Sayad Noveiri, Roghayeh Yaghoobi Saghezchi, Mohammad Javad Jafari, Reza Nemati-Vakilabad.

**Formal analysis:** Alireza Mirzaei, Reza Imashi, Reza Nemati-Vakilabad.

**Investigation:** Mobina Jamshidinia, Marzieh Jahani Sayad Noveiri, Reza Imashi, Mohammad Javad Jafari, Reza Nemati-Vakilabad.

**Methodology:** Alireza Mirzaei, Marzieh Jahani Sayad Noveiri, Reza Imashi, Roghayeh Yaghoobi Saghezchi, Reza Nemati-Vakilabad.

**Project administration:** Roghayeh Yaghoobi Saghezchi, Reza Nemati-Vakilabad.

**Validation:** Mobina Jamshidinia, Marzieh Jahani Sayad Noveiri, Reza Imashi, Reza Nemati-Vakilabad.

**Visualization:** Roghayeh Yaghoobi Saghezchi.

**Writing – original draft:** Alireza Mirzaei, Mobina Jamshidinia, Marzieh Jahani Sayad Noveiri, Reza Imashi, Roghayeh Yaghoobi Saghezchi, Mohammad Javad Jafari, Reza Nemati-Vakilabad.

**Writing – review & editing:** Alireza Mirzaei, Mobina Jamshidinia, Marzieh Jahani Sayad Noveiri, Reza Imashi, Roghayeh Yaghoobi Saghezchi, Mohammad Javad Jafari, Reza Nemati-Vakilabad.

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
