## [Decision Letter · Decision Letter 0]

7 Apr 2025

Dear Dr. Nemati-Vakilabad,

Thank you for submitting your manuscript to PLOS ONE. After careful consideration, we feel that it has merit but does not fully meet PLOS ONE’s publication criteria as it currently stands. Therefore, we invite you to submit a revised version of the manuscript that addresses the points raised during the review process.

We look forward to receiving your revised manuscript.

Kind regards,

Noushin Kohan

Academic Editor

PLOS ONE

**Journal Requirements:**

1. When submitting your revision, we need you to address these additional requirements. Please ensure that your manuscript meets PLOS ONE's style requirements, including those for file naming. The PLOS ONE style templates can be found at https://journals.plos.org/plosone/s/file?id=wjVg/PLOSOne_formatting_sample_main_body.pdf and https://journals.plos.org/plosone/s/file?id=ba62/PLOSOne_formatting_sample_title_authors_affiliations.pdf 2. In the online submission form, you indicated that The datasets used and analyzed during the current study are available from the corresponding author upon reasonable request. All PLOS journals now require all data underlying the findings described in their manuscript to be freely available to other researchers, either a. In a public repository, b. Within the manuscript itself, or c. Uploaded as supplementary information.This policy applies to all data except where public deposition would breach compliance with the protocol approved by your research ethics board. If your data cannot be made publicly available for ethical or legal reasons (e.g., public availability would compromise patient privacy), please explain your reasons on resubmission and your exemption request will be escalated for approval. 3. We note that this data set consists of interview transcripts. Can you please confirm that all participants gave consent for interview transcript to be published? If they DID provide consent for these transcripts to be published, please also confirm that the transcripts do not contain any potentially identifying information (or let us know if the participants consented to having their personal details published and made publicly available). We consider the following details to be identifying information:- Names, nicknames, and initials- Age more specific than round numbers- GPS coordinates, physical addresses, IP addresses, email addresses- Information in small sample sizes (e.g. 40 students from X class in X year at X university)- Specific dates (e.g. visit dates, interview dates)- ID numbers Or, if the participants DID NOT provide consent for these transcripts to be published:- Provide a de-identified version of the data or excerpts of interview responses- Provide information regarding how these transcripts can be accessed by researchers who meet the criteria for access to confidential data, including:a) the grounds for restrictionb) the name of the ethics committee, Institutional Review Board, or third-party organization that is imposing sharing restrictions on the datac) a non-author, institutional point of contact that is able to field data access queries, in the interest of maintaining long-term data accessibility.d) Any relevant data set names, URLs, DOIs, etc. that an independent researcher would need in order to request your minimal data set. For further information on sharing data that contains sensitive participant information, please see: https://journals.plos.org/plosone/s/data-availability#loc-human-research-participant-data-and-other-sensitive-data If there are ethical, legal, or third-party restrictions upon your dataset, you must provide all of the following details (https://journals.plos.org/plosone/s/data-availability#loc-acceptable-data-access-restrictions):a) A complete description of the datasetb) The nature of the restrictions upon the data (ethical, legal, or owned by a third party) and the reasoning behind themc) The full name of the body imposing the restrictions upon your dataset (ethics committee, institution, data access committee, etc)d) If the data are owned by a third party, confirmation of whether the authors received any special privileges in accessing the data that other researchers would not havee) Direct, non-author contact information (preferably email) for the body imposing the restrictions upon the data, to which data access requests can be sent?

**Additional Editor Comments:**

This article is publishable with minor revisions. The study provides valuable insights into dignity measurement in healthcare settings and offers a validated tool that can be used for improving patient care. However, some areas require clarification or elaboration to enhance the manuscript's quality and impact:

Reviewers' comments:

Reviewer's Responses to Questions

**Comments to the Author**

1. Is the manuscript technically sound, and do the data support the conclusions?

Reviewer #1: Yes

Reviewer #2: Yes

2. Has the statistical analysis been performed appropriately and rigorously?

Reviewer #1: Yes

Reviewer #2: Yes

3. Have the authors made all data underlying the findings in their manuscript fully available?

Reviewer #1: Yes

Reviewer #2: Yes

4. Is the manuscript presented in an intelligible fashion and written in standard English?

Reviewer #1: Yes

Reviewer #2: Yes

**Reviewer #1:**  Page 6 Translation procedure, it may be useful to put in the citations for the HOADS by Dr. Abdul-Ganiyu Fuseini, and the citation for the WHO guidelines.

On page 7, Content validity, were the same panel of experts used in the qualitative assessment used for the quantitative evaluation?

**Reviewer #2: ** The discussion section could provide more comparative insights with similar psychometric validation studies in other languages or settings.

A brief explanation of potential limitations (e.g., cultural biases in self-reported dignity assessments) would enhance the study’s credibility.

Future research directions could be elaborated further, including potential applications of the HOADS-P in different healthcare settings beyond acute hospitalization.

**Do you want your identity to be public for this peer review?** For information about this choice, including consent withdrawal, please see our Privacy Policy

Reviewer #1: **Yes: ** Carlos Diego A. Rozul

Reviewer #2: No

---

## [Author Response · Author response to Decision Letter 1]

12 Apr 2025

Dear Editorial Board of PLOS ONE,

We sincerely appreciate the opportunity to submit our revised manuscript to PLOS ONE. We would like to express our gratitude to the academic editor and reviewers for their time and constructive feedback, which has significantly strengthened our work.

Please find our point-by-point responses to all reviewer comments, along with all requested modifications highlighted in the revised manuscript. We have carefully addressed each concern and believe the manuscript now meets the journal's high standards.

Should any additional clarifications be needed, we would be pleased to provide them. Thank you for your consideration of our work.

With best regards,

Reza Nemati-Vakilabad

---

## [Decision Letter · Decision Letter 1]

26 Aug 2025

Psychometric properties of the Persian version of the hospitalized older adults' dignity scale for measuring dignity during acute hospitalization

PONE-D-25-03918R1

Dear Dr. Reza Nemati-Vakilabad,

We’re pleased to inform you that your manuscript has been judged scientifically suitable for publication and will be formally accepted for publication once it meets all outstanding technical requirements.

Kind regards,

Daniel Chukwuemeka Ogbuabor, Ph.D., M.D.

Academic Editor

PLOS ONE

Additional Editor Comments (optional):

Reviewers' comments:

Reviewer's Responses to Questions

**Comments to the Author**

Reviewer #1: All comments have been addressed

Reviewer #2: All comments have been addressed

2. Is the manuscript technically sound, and do the data support the conclusions?

Reviewer #1: Yes

Reviewer #2: Yes

3. Has the statistical analysis been performed appropriately and rigorously?

Reviewer #1: Yes

Reviewer #2: Yes

4. Have the authors made all data underlying the findings in their manuscript fully available?

Reviewer #1: Yes

Reviewer #2: Yes

5. Is the manuscript presented in an intelligible fashion and written in standard English?

Reviewer #1: Yes

Reviewer #2: Yes

Reviewer #1: All reviewer comments have been addressed. This is an excellent paper that will surely be of use for many healthcare workers.

Reviewer #2: The authors have adequately addressed all the comments raised during the previous review round. The manuscript is now clear, technically sound, and methodologically rigorous. The psychometric analyses are well conducted, with appropriate validity and reliability measures. Ethical approval and informed consent procedures are clearly stated, and the limitations section has been appropriately expanded. The manuscript is well-written in standard English, and the presentation is coherent and accessible.

I have no further major concerns, and I recommend the manuscript for publication in its current form.

**Do you want your identity to be public for this peer review?** For information about this choice, including consent withdrawal, please see our Privacy Policy

Reviewer #1: **Yes: ** Carlos Diego A. Rozul

Reviewer #2: **Yes: ** Dr Gihan Mohamed Mohamed Salem

---

## [Editor Report · Acceptance letter]

PONE-D-25-03918R1

PLOS ONE

Dear Dr. Nemati-Vakilabad,

I'm pleased to inform you that your manuscript has been deemed suitable for publication in PLOS ONE. Congratulations! Your manuscript is now being handed over to our production team.

Kind regards,

on behalf of

Dr. Daniel Chukwuemeka Ogbuabor

Academic Editor

PLOS ONE